# Tamoxifen and Fertility in Women with Breast Cancer: A Systematic Review on Reproductive Outcomes and Oncological Safety of Treatment Interruption

**DOI:** 10.3390/ijms26083787

**Published:** 2025-04-17

**Authors:** Mauro Francesco Pio Maiorano, Gennaro Cormio, Vera Loizzi, Brigida Anna Maiorano, Stella D’Oronzo, Erica Silvestris

**Affiliations:** 1Gynecologic Oncology Unit, IRCCS Istituto Tumori “Giovanni Paolo II”, 70124 Bari, Italy; gennaro.cormio@uniba.it (G.C.); vera.loizzi@uniba.it (V.L.); e.silvestris@oncologico.bari.it (E.S.); 2Department of Translational Biomedicine and Neuroscience (DiBraiN), University of Bari “Aldo Moro”, Policlinico of Bari, Piazza Giulio Cesare 11, 70124 Bari, Italy; 3Department of Medical Oncology, IRCCS San Raffaele Hospital, Via Olgettina 60, 20132 Milan, Italy; maiorano.brigida@hsr.it; 4Medicine and Surgery Department, LUM University, Casamassima, 70010 Bari, Italy; 5Oncology and Oncohematology Division, “F. Miulli” General Regional Hospital, Acquaviva delle Fonti, 70021 Bari, Italy

**Keywords:** tamoxifen breast cancer, tamoxifen therapy, breast cancer fertility preservation, breast cancer recurrence, tamoxifen pregnancy success, tamoxifen pregnancy side effects, breast cancer pregnancy, tamoxifen interruption for pregnancy, systematic review tamoxifen fertility, breast cancer survival

## Abstract

Breast cancer (BC) is the most prevalent malignancy among women worldwide, with a rising incidence in young, premenopausal patients. For those diagnosed with hormone receptor-positive (HR+) BC, tamoxifen is a cornerstone of adjuvant endocrine therapy, significantly reducing recurrence risk and improving long-term survival. However, its prolonged use poses challenges for women desiring pregnancy, prompting interest in temporary treatment interruption as a strategy to achieve reproductive goals while maintaining oncological safety. This systematic review evaluates the impact of tamoxifen on fertility, the feasibility of treatment interruption, and associated reproductive and oncological outcomes. Following the Preferred Reporting Items for Systematic Reviews and Meta-Analyses (PRISMA) guidelines, we conducted a comprehensive search across major databases, identifying three relevant studies, including one randomized controlled trial (RCT) and two observational cohort studies. The findings suggest that temporary tamoxifen interruption allows for successful pregnancies without significantly increasing short-term recurrence rates. Notably, the POSITIVE trial demonstrated a pregnancy achievement rate of 74% and a live birth rate of 63.8%, with comparable three-year disease-free survival between patients who interrupted tamoxifen and those who continued therapy. However, concerns remain regarding tamoxifen’s teratogenic risks, emphasizing the need for strict contraceptive measures and preconception counseling. Despite emerging evidence supporting this approach, long-term safety data are limited. Further research is warranted to refine clinical recommendations and optimize reproductive counseling for young BC survivors.

## 1. Introduction

Breast cancer (BC) is the most common malignancy affecting women worldwide, with an increasing incidence in reproductive age, particularly between 30 and 45 years (yrs) [1]. Approximately around 4% of the invasive BC events occur in women under 40 yrs [2]. Between 2017 and 2021, it has been estimated that the yearly age-adjusted rate of new cases of BC in women was 129.4 for every 100,000 individuals [3]. Approximately 100 of these cases are represented by early-stage BC diagnosis [3]. Encouragingly, advancements in early detection and treatment have led to a decline in age-adjusted death rates by 1.2% annually from 2013 to 2022, resulting in a 5-year relative survival rate of 91.2% in 2020 [4]. Among pre-menopausal women diagnosed with BC, the 5-year survival rate is 88%, compared to 91% for those aged 45–54 yrs and 92% for those aged 65–74 yrs [3,4]. Recent advances in diagnosis and therapy have significantly improved BC survival rates, turning attention to quality-of-life issues, including fertility preservation (FP). Fertility preservation refers to several medical interventions aimed at safeguarding reproductive potential before or during cancer treatment, such as the cryopreservation of oocytes, embryos, or ovarian tissue, as well as the use of gonadotropin-releasing hormone (GnRH) analogs to reduce ovarian damage during gonadotoxic treatments [5]. These practices are particularly significant for pre-menopausal women who experience fertility loss due to oncological treatments, as many wish to conceive after recovering from cancer. Among BC molecular subtypes, hormone-receptor-positive (HR+) disease holds particular relevance in this context, given its prevalence among young women [6]. Notably, HR+ BC accounts for approximately 70% of all BC cases in the pre-menopausal population, further emphasizing the need to address fertility-related concerns in these patients [7]. For such patients, tamoxifen is a cornerstone of adjuvant endocrine therapy, which may additionally involve aromatase inhibitors (in postmenopausal women), and ovarian function suppression in combination with tamoxifen or aromatase inhibitors for high-risk premenopausal patients [8,9,10,11]. Tamoxifen, a selective estrogen receptor modulator (SERM), binds to estrogen receptors, exerting an antagonist effect in breast tissue while acting as an agonist in bone and uterine tissues. It is typically prescribed for 5 to 10 years in the adjuvant setting and has been shown to significantly reduce recurrence risk and mortality by up to 31% with five years of treatment [12]. However, its prolonged use creates a significant challenge for women desiring pregnancy [13]. The gonadotoxic effects of systemic therapies can lead to premature ovarian insufficiency (POI) and impaired fertility, and it is important to note that tamoxifen’s teratogenicity requires strict contraception during treatment and appropriate washout periods after its conclusion [14,15]. Moreover, many women who complete the recommended course of tamoxifen therapy inevitably reach an age at which natural fertility has physiologically declined, further complicating their chances of conception. This age-related failure of ovarian reserve, combined with the potential impact of prior cancer treatments, underscores the importance of timely FP strategies and individualized reproductive counseling. Current guidelines recommend a washout period of at least two months before conception to minimize the risk of congenital anomalies, based on tamoxifen’s half-life [16]. Some experts suggest three months to minimize more potential risks [15]. For women planning pregnancy, temporary interruption of hormonal therapy, including tamoxifen, is increasingly considered a viable option despite the lack of extensive evidence [17]. This approach allows patients to attempt conception before resuming therapy, particularly for those in an age group where completing the full course of treatment would significantly limit their reproductive window. However, while emerging evidence, including data from the POSITIVE trial, suggests that treatment interruption may be oncologically safe in carefully selected patients, the long-term safety of this approach remains still uncertain due to the scarcity of strong data [17]. The balance between ensuring optimal oncological outcomes and preserving fertility is complex, highlighting the need for individualized decision-making and further research to guide clinical recommendations. This systematic review aims to present current literature evidence concerning the impact of tamoxifen employed in the treatment of pre-menopausal BC patients on long-term fertility and pregnancy desire. It also seeks to evaluate the safety of treatment interruption for conception and the resulting reproductive and fetal outcomes. Finally, by addressing these crucial questions, this review aims to inform clinical decision-making and promote a patient-centered approach that integrates fertility goals with cancer care.

## 2. Materials and Methods

This systematic review was conducted to evaluate the impact of tamoxifen on fertility, the safety of treatment interruption for pregnancy, and the associated reproductive outcomes in pre-menopausal HR+ BC women. The review followed the Preferred Reporting Items for Systematic Reviews and Meta-Analyses (PRISMA) guidelines [18]. We registered the protocol for this systematic review with PROSPERO (CRD42024619107). The inclusion and exclusion criteria were defined using the PICOS framework [19]. The population (P) of interest included pre-menopausal women diagnosed with HR+ BC undergoing adjuvant tamoxifen therapy, with subgroups including those who paused tamoxifen therapy to conceive, those who completed therapy, and those exposed to tamoxifen during pregnancy. The analyzed interventions (I) were tamoxifen therapy as standard adjuvant endocrine treatment, pregnancy achieved after tamoxifen therapy, and temporary interruption of tamoxifen for pregnancy. Comparators (C) included women enduring uninterrupted tamoxifen therapy, those not achieving pregnancy after tamoxifen, and those not undergoing tamoxifen treatment. The primary outcomes assessed were reproductive events such as pregnancy rates, live birth rates, and ovarian reserve evaluation, including anti-Müllerian hormone (AMH) sieric levels and the incidence of premature ovarian insufficiency (POI). Secondary outcomes included oncological safety parameters such as recurrence rates, disease-free survival (DFS), overall survival (OS), pregnancy-related complications, and fetal outcomes. Studies eligible for inclusion (S) were randomized controlled trials (RCTs), prospective or retrospective cohort studies, and observational studies reporting these outcomes. Case reports, narrative reviews, and studies lacking reproductive or oncological outcomes were excluded. The search was limited to English-written studies. The PICOS structure for study selection is summarized in Table 1.

A comprehensive search was conducted across PubMed, EMBASE, and the Cochrane Library for studies published up to 31 December 2024. The following keywords and Boolean operators were used: “tamoxifen” AND (“breast cancer” OR “breast carcinoma”) AND (“fertility preservation” OR “pregnancy outcomes” OR “treatment interruption”). The reference lists of selected studies and related reviews were also screened to identify additional eligible articles. The database search identified 1500 studies. After removing 298 duplicates, 1200 unique records were screened based on titles and abstracts. Of these, 900 records were excluded for not meeting the eligibility criteria, leaving 300 full-text articles for assessment. Following a detailed review, 297 articles were excluded for irrelevance, lack of reproductive outcomes, or missing data. Three studies were ultimately included in this review. The PRISMA flow chart summarizing the selection process is presented in Figure 1.

Two reviewers, M.F.P.M. and E.S., independently extracted data using a pre-defined template. The extracted data included study characteristics, population demographics, intervention details, comparator information, and outcomes. Discrepancies were resolved through consensus, and a third reviewer adjudicated unresolved conflicts. The Cochrane Risk of Bias (RoB 2) tool was used for RCTs, while the Newcastle–Ottawa Scale (NOS) was applied to observational studies. The global risk of bias for the selected studies was low in the only RCT, and low-to-moderate for the observational and retrospective cohorts of the other studies (Appendix A).

## 3. Results

A total of 974 women were included across three studies: one RCT and two observational cohort studies [13,17,20]. Table 2 summarizes the characteristics of the included studies.

### 3.1. Findings from the POSITIVE Trial

The POSITIVE trial was a phase III RCT that included 516 pre-menopausal women with previous stage I–III HR+ BC receiving tamoxifen for 18 to 30 months. The primary endpoint was the incidence of BC adverse events during the follow-up period, which included local, regional, or distant recurrences of invasive BC, as well as the occurrence of new invasive BC on the contralateral side. After a median follow-up of 41 months, 263 patients temporarily suspended tamoxifen administration (group 1) to attempt pregnancy, and 253 continued therapy without interruption (group 2). In the first group, the pregnancy achievement rate was 74%, with a live birth rate of 63.8%. No significant differences in three-year recurrence rates were reported between the two groups (8.9% vs. 9.2%). The 3-year incidence of distant recurrences was 4.5% (95% CI, 2.7–6.4) in the tamoxifen interruption group and 5.8% (95% CI, 4.5–7.2) in the control group. This represented an absolute difference of −1.4 percentage points (95% CI, −3.5 to 1.0), calculated using the bootstrap-matching method. The hazard ratio (HR) for distant recurrence in group 1 compared to group 2 was 0.70 (95% CI, 0.44–1.12). The study protocol required a three-month washout period between halting endocrine therapy and beginning attempts to conceive. Birth defects were reported to be 2.2% among the 365 offspring, in line with the rates observed in the general population [17].

### 3.2. Findings from Shandley et al.

Shandley et al. analyzed a retrospective cohort of 397 pre-menopausal BC survivors aged 20–45 years, including 179 tamoxifen users and 218 non-users, with a median of 7 years from the diagnosis. Patients were defined as “tamoxifen users” if exposed to endocrine therapy for at least 6 months. The live birth rate was significantly lower among tamoxifen users, with an HR of 0.25 (95% CI: 0.14–0.47). However, tamoxifen users exhibited higher ovarian reserve markers, including median anti-Müllerian hormone (AMH) levels of 2.3 ng/mL versus 1.8 ng/mL in non-users, and median antral follicle count (AFC) of 15 versus 11 follicles [13].

### 3.3. Findings from Nye et al.

Nye et al. followed 61 premenopausal women who had completed tamoxifen therapy, proving that the exposure to endocrine therapy was significantly longer (*p* = 0.008) in the no-pregnancy cohort, a median of 42.3 months (range 0–120 mos), compared to the pregnancy cohort (median 20.9 months, range 0–72 mos). Among the 31 women who pursued pregnancy, the median time from tamoxifen cessation to conception was 7 months, and the live birth rate was 77%. The follow-up duration was significantly shorter in the control group (no pregnancy) compared to the pregnancy group, with a mean follow-up of 78 months (range: 23–168 months) versus 110 months (range: 57–185 months), respectively (*p* = 0.005). There was no statistically significant difference in BC recurrence between women who became pregnant within five years of their breast cancer diagnosis and those who did not. Recurrence occurred in four women (14%) in the control group and eight women (26%) in the pregnancy group (*p* = 0.34). The five-year DFS was 92% (95% CI: 81–100%) for women who did not become pregnant and 84% (95% CI: 72–97%) for women who achieved pregnancy within five years of diagnosis. No statistically significant difference in five-year DFS was observed between the two groups (*p* = 0.69) [20].

## 4. Discussion

We provided the first systematic review exploring the impact of tamoxifen on fertility and the safety of treatment interruption for BC patients desiring pregnancy.

### 4.1. Teratogenic Risks of Tamoxifen

Our study proves that tamoxifen therapy’s temporary interruption allows for successful pregnancies without compromising short-term oncological outcomes. However, tamoxifen exposure during pregnancy is associated with significant teratogenic risks, emphasizing the need for careful contraceptive management and preconception planning [20]. The teratogenic risks of tamoxifen exposure during pregnancy are a critical consideration for pre-menopausal women with BC who wish to conceive. Although several studies included in this systematic review provide strong evidence of pregnancy success rates and oncological safety following tamoxifen interruption, additional insights can be drawn from observational data and comprehensive reviews of real-world cases [13,17,20]. Braems et al. analyzed 417 pregnancies exposed to tamoxifen, reporting a 13.1% incidence of congenital anomalies among 138 live births. These anomalies predominantly involved craniofacial and skeletal malformations. Additionally, spontaneous abortions occurred in 14% of cases, and therapeutic terminations were documented in 6.7% of cases [16]. These findings underscore the teratogenic risks of tamoxifen, particularly when conception occurs without an adequate washout period. The study recommended a minimum two-month washout period before conception to reduce fetal risks, a widely adopted precaution in clinical practice [16]. In support of these results, Schuurman and colleagues summarized data from 238 cases of tamoxifen exposure during pregnancy reported in the literature. Their review highlighted similar concerns regarding congenital anomalies, spontaneous abortions, and preterm deliveries, noting that the overall incidence of fetal abnormalities ranged from 10% to 20% [21]. Tamoxifen, in fact, as a selective estrogen receptor modulator can cross the placental barrier, interfering with estrogen-dependent developmental processes. This mechanism likely underlies the observed craniofacial and urogenital anomalies reported in tamoxifen-exposed pregnancies [16,21,22,23]. These findings further emphasize the importance of rigorous contraceptive measures during tamoxifen therapy and careful planning for pregnancy attempts.

### 4.2. Psychosocial Consideration and Fertility Planning

The review also stressed the need for comprehensive counseling to address the psychological and ethical complexities associated with these cases. The emotional burden of balancing cancer treatment with reproductive planning is considerable. Fear of recurrence, delayed parenthood, and concerns about fertility loss contribute to significant psychological distress. In this context, mental health professionals and specialized oncofertility teams play a vital role in supporting patients through complex and emotionally charged decisions. Biological mechanisms underlying the oncological safety of pregnancy may also provide additional reassurance for patients and clinicians. Pregnancy-induced hormonal changes, such as elevated levels of progesterone and alterations in estrogen metabolism, could reduce the proliferative potential of residual HR+ BC cells [24,25,26]. Progesterone, in particular, may exert a protective effect by promoting differentiation in mammary epithelial cells [27]. Furthermore, pregnancy is associated with immunomodulatory shifts, including increased activity of regulatory T cells and natural killer cells, which might contribute to a tumor-suppressive microenvironment [28,29,30]. While promising, these mechanisms remain speculative and require further investigation to confirm their clinical implications. Jointly, our findings provide critical insights into balancing cancer treatment with fertility and pregnancy goals. These results highlight the complex interaction between tamoxifen therapy, fertility preservation, and pregnancy outcomes in pre-menopausal women with HR+ BC. The role of tamoxifen in reducing BC recurrence and mortality in HR+ early-stage BC is well established. The Early Breast Cancer Trialists’ Collaborative Group (EBCTCG) meta-analysis demonstrated that five years of tamoxifen reduces annual BC mortality by 31% and recurrence risk by nearly 50% compared to no treatment, with benefits persisting even after treatment cessation [12]. These findings support current guidelines recommending five to ten years of adjuvant tamoxifen therapy, particularly for high-risk patients [11]. However, this prolonged treatment duration conflicts with reproductive timelines for many young women. In clinical practice, the decision to interrupt tamoxifen for pregnancy requires comprehensive counseling that goes beyond oncological risk assessment. Patients must be informed not only of the relative safety of temporary discontinuation but also of the nuanced trade-offs involving fertility timelines, ovarian reserve, and the psychological burden of pausing therapy. Emerging frameworks for oncofertility counseling highlight the value of integrating reproductive endocrinologists early in the decision-making process to optimize fertility preservation planning and ensure appropriate contraceptive strategies during and after treatment. The use of reproductive decision aids, fertility navigators, and structured follow-up protocols may help facilitate shared decision-making and improve long-term satisfaction with care. Multidisciplinary collaboration is essential to tailor treatment interruption windows to both oncologic risk and reproductive potential, especially for women nearing the end of their fertile window.

### 4.3. Oncological Safety of Tamoxifen Interruption

The POSITIVE trial provides crucial evidence that tamoxifen interruption for pregnancy attempts can be both feasible and oncologically safe. With 263 participants temporarily halting therapy for up to two years to conceive, the study demonstrated high pregnancy success rates and live birth outcomes, without significantly increasing the risk of short-term recurrence [17]. Furthermore, Shandley et al. highlighted the lower live birth rates among tamoxifen users compared to non-users, although remarking higher ovarian reserve markers in the tamoxifen group, which may suggest a protective effect on ovarian function [13]. Furthermore, concerns about future fertility may influence treatment adherence, with some women delaying or discontinuing tamoxifen due to reproductive goals [13]. This underscores the importance of integrating fertility counseling at the time of diagnosis to align oncological safety with patients’ reproductive desires and to support adherence to treatment plans. While tamoxifen remains the cornerstone of endocrine therapy in young women with HR+ BC, it is often administered in combination with ovarian function suppression (OFS) or chemotherapy, especially in high-risk or node-positive cases. OFS, achieved via GnRH agonists or oophorectomy, leads to temporary or permanent amenorrhea, depending on patient age and duration. Chemotherapy, particularly alkylating agents, is known to be gonadotoxic, significantly impacting fertility through loss of ovarian reserve. In fact, Goldfarb et al. conducted a study comparing ovarian reserve among patients undergoing different breast cancer treatments. They found that chemotherapy regimens led to a significant decline in ovarian reserve, as evidenced by reduced AMH levels. In contrast, patients receiving tamoxifen-only treatment did not experience a significant alteration in ovarian reserve over a 24-month follow-up period [31]. These findings suggest that while tamoxifen use is associated with lower live birth rates, it does not adversely affect, and may even preserve, ovarian reserve markers such as AMH and AFC. This preservation could be due to tamoxifen’s selective estrogen receptor modulation, which might protect ovarian function during treatment and may also be mediated through tamoxifen’s anti-estrogenic feedback at the hypothalamic–pituitary level, i.e., reducing gonadotropin stimulation and follicular depletion [13,31]. However, further research is necessary to fully understand the mechanisms underlying these observations and their clinical implications. Nye et al. reinforced the oncological safety of pregnancy after tamoxifen therapy, with no substantial differences in disease-free survival or recurrence rates between women who conceived and those who did not [20]. These findings collectively highlight the importance of individualized treatment planning and support the feasibility of integrating reproductive goals into BC care. The temporal assumption of tamoxifen therapy before interruption is also a critical factor influencing patient outcomes. The POSITIVE trial emphasized the importance of completing at least 18 to 24 months of therapy before attempting interruption, but questions regarding this timing remain open and should be confirmed by more and larger studies involving diverse population [17]. This aligns with evidence suggesting that early tamoxifen interruption (before two years) may increase BC recurrence risk by up to 20–25% compared to completing the full five-year course [32,33]. A visual summary of the treatment timeline and decision-making process for fertility considerations in young women undergoing tamoxifen therapy is presented in Figure 2. This diagram highlights the clinical rationale, potential risks, and benefits of hormonal therapy interruption and illustrates the current evidence supporting its oncological safety based on the included trials while also emphasizing the persistent gap in knowledge regarding the optimal timing, the duration of treatment interruption, and its validation in larger cohort studies.

### 4.4. Breast Cancer Patients and ART

Assisted reproductive technologies (ART), such as IVF and embryo cryopreservation, offer valuable fertility options for breast cancer survivors. While ART are generally considered safe, data remain limited on their long-term efficacy and safety following tamoxifen therapy in hormone-sensitive patients. Individualized risk assessments and careful endocrinologic monitoring are essential to guide ART decisions in this population [13,34]. We believe that fertility preservation strategies involving ART, such as oocyte or embryo cryopreservation, are essential for young women with BC. Tamoxifen users often demonstrate higher ovarian reserve markers, which may improve the success rates of fertility preservation [11,13,30]. Recent studies have explored the use of tamoxifen in controlled ovarian hyperstimulation protocols for fertility preservation, suggesting its potential efficacy in this setting [34,35,36,37]. Some evidence suggests that tamoxifen might not adversely affect ovarian reserve and may even be associated with increased AMH levels in premenopausal women undergoing treatment [13]. However, the underlying mechanisms are not fully understood, and further research is necessary to determine whether this reflects a genuine protective effect or is influenced by confounding factors such as patient age or baseline ovarian reserve. Addressing the psychosocial impact of fertility preservation and pregnancy in cancer survivors is critical to ensuring holistic care [38]. Access to oncofertility services varies widely across regions, with significant disparities in low- and middle-income countries [39]. Addressing these inequities requires policy-level interventions to ensure that all patients, regardless of socioeconomic status, can access FP and reproductive health services [40].

### 4.5. Limitations and Future Directions

Despite promising results, several gaps in knowledge remain. There is a lack of high-quality data on the impact of tamoxifen interruption in diverse populations, especially those from low- and middle-income countries where access to fertility preservation and cancer care is limited. Moreover, there is little information on the safety of repeated pregnancy attempts or fertility interventions following an initial pregnancy post-tamoxifen interruption. Future prospective studies should investigate not only long-term oncologic outcomes but also patient-reported outcomes, including reproductive satisfaction, anxiety, and decisional regret. Incorporating molecular and genomic profiling could also help identify which patients might benefit most from treatment interruption with minimal oncologic risk. As reproductive goals gain recognition in survivorship care plans, these future investigations will be crucial in evolving standards of care for young women with HR+ BC. Furthermore, our systematic review has some limitations that should be addressed. First, despite the POSITIVE trial providing strong evidence, its follow-up duration does not involve long-term recurrence and survival outcomes [17]. Moreover, the other included studies are observational, introducing selection risks and reporting biases. Future research should explore the long-term safety of tamoxifen interruption, the effects on subsequent pregnancies, and the optimal timing for therapy resumption. Furthermore, some of the reported studies had small sample sizes, which reduced the statistical power to detect subtle differences or rare events, such as congenital anomalies or long-term outcomes. Variability in the duration of tamoxifen therapy before the interruption, the timing of the interruption, and the subsequent resumption of treatment, which were not standardized among the studies, also complicate the ability to draw definitive conclusions about the optimal timing and duration of therapy interruption. Additionally, while this systematic review provides valuable information concerning the outcomes of first pregnancies following tamoxifen therapy, it offers limited data on subsequent pregnancies, which are important considerations for many BC survivors. Finally, the exclusion of unpublished data may have introduced publication bias, potentially overlooking relevant findings. Addressing these limitations would require high-quality RCTs with longer follow-up periods, larger and more diverse populations, and uniform criteria for tamoxifen interruption and resumption. Incorporating patient-reported outcomes and psychosocial assessments would also provide a more comprehensive understanding of the impact of these interventions on quality of life.

## 5. Conclusions

This systematic review underscores the critical junction of tamoxifen therapy, reproductive goals, and oncological safety in young women with BC/HR+. The studies reviewed demonstrate that a temporary interruption of tamoxifen can be a feasible option for carefully selected patients who wish to conceive, with high pregnancy success rates and comparable short-term recurrence outcomes [13,17,20]. Furthermore, evidence of tamoxifen’s potential ovarian-protective effects adds a nuanced perspective to its role in fertility preservation strategies. However, despite these promising results, significant challenges remain, particularly in addressing teratogenic risks, optimizing timing for therapy interruption, and improving access to fertility preservation services globally. Long-term follow-up data and larger-scale studies are necessary to ensure the safety and efficacy of tamoxifen interruption over extended periods. Furthermore, the integration of psychosocial and economic factors into patient care can improve decision-making and support management approaches, highlighting the need for multidisciplinary collaboration in the evolving field of oncofertility.

## Figures and Tables

**Figure 1 ijms-26-03787-f001:**
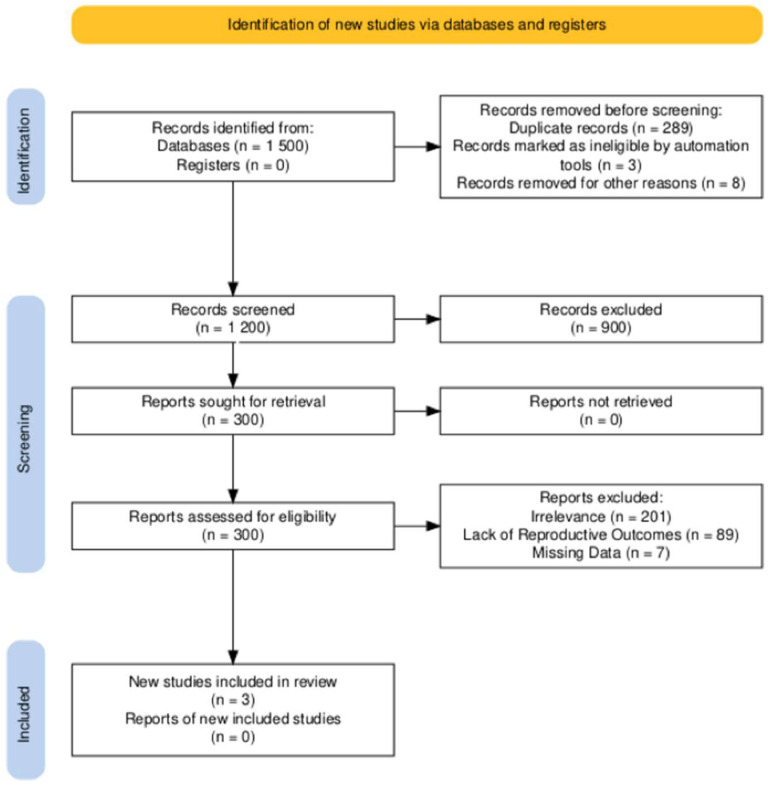
PRISMA flowchart of study selection.

**Figure 2 ijms-26-03787-f002:**
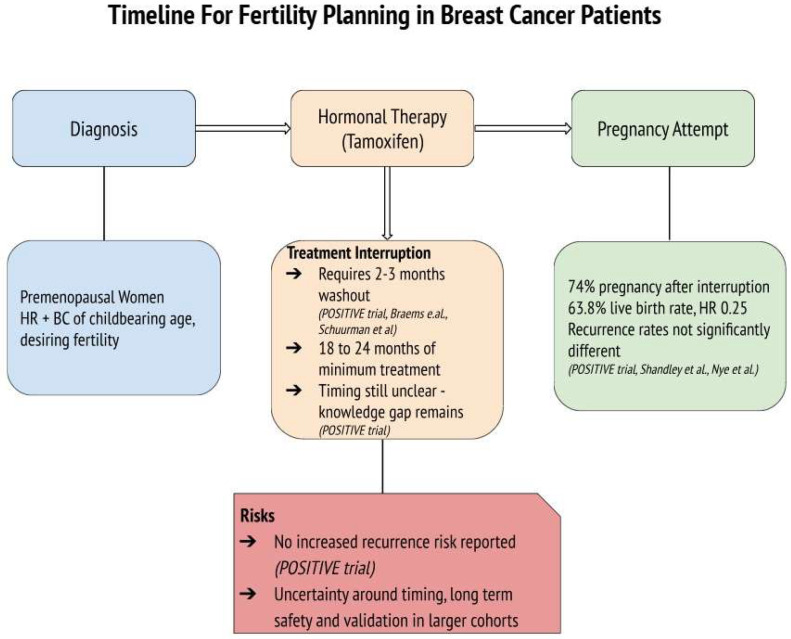
Timeline for fertility planning in premenopausal women with hormone receptor-positive (HR+) breast cancer undergoing tamoxifen therapy. The diagram outlines key steps and considerations for treatment interruption in patients desiring pregnancy, reporting the main findings from the studies [13,16,17,20,21].

**Table 1 ijms-26-03787-t001:** PICOS criteria for studies selected for the systematic review.

**Population**	Premenopausal women with non-metastatic HR+ BC undergoing or completed tamoxifen therapy
**Intervention**	Tamoxifen, pregnancy under tamoxifen, or its temporary interruption to attempt pregnancy
**Control**	Tamoxifen non-users, tamoxifen continuation, no pregnancy under tamoxifen
**Outcomes**	Pregnancy rates, live birth rates, ovarian reserve (i.e., AMH, AFC), recurrence rates, DFS, OS, pregnancy-related and fetal outcomes
**Studies**	RCTs and observational cohort studies

AMH, anti-Müllerian hormone; AFC, antral follicle count; BC, breast cancer; DFS, disease-free survival; HR+, hormone-receptor-positive; OS, overall survival; RCT, randomized controlled trial.

**Table 2 ijms-26-03787-t002:** Summary of included studies.

Study (First Author, Year)	Population and Comparator	Intervention	Outcomes	Key Findings
**Partridge et al., 2023 (POSITIVE) [17]**	516 pre-menopausal women (≤42 years) with HR+ BC: 263 interrupted tamoxifen to achieve pregnancy vs. 253 continued	Tamoxifen interruption (18–30 mos of prior use)	74% achieved pregnancy; 63.8% live birth rate; 3-year recurrence rate: 8.9% (vs. 9.2%)	Safe temporary interruption for pregnancy without increased short-term BC recurrence risk
**Shandley et al., 2017 [13]**	397 pre-menopausal HR+ BC survivors: 179 tamoxifen users vs. 218 non-users (20–45 years)	Tamoxifen therapy (≥6 mos)	HR for live birth in tamoxifen users: 0.25 (95% CI: 0.14, 0.47); AMH and AFC higher in tamoxifen users vs. non-users	Tamoxifen linked to fewer post-diagnosis births but no significant ovarian reserve reduction
**Nye et al., 2017 [20]**	61 pre-menopausal women with HR+ BC: 31 became pregnant after therapy, 30 did not attempt pregnancy	Pregnancy after tamoxifen: control (no pregnancy) cohort, 42.3 (0–120) months, pregnancy cohort with a mean of 20.9 (0–72) months (*p* = 0.008)	DFS: 84% (pregnancy cohort) vs. 92% (control); recurrence not significantly different	Pregnancy after tamoxifen therapy did not worsen BC outcomes

AMH, anti-Müllerian hormone; AFC, antral follicle count; BC, breast cancer; DFS, disease-free survival; HR+, hormone-receptor-positive.

## Data Availability

The authors confirm that the data supporting the findings of this study are available within the article and its Appendix A.

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
