# Peer review of "Tamoxifen and Fertility in Women with Breast Cancer: A Systematic Review on Reproductive Outcomes and Oncological Safety of Treatment Interruption"

_ijms, 2025, doi:10.3390/ijms26083787_

Round 1
Reviewer 1 Report
Comments and Suggestions for Authors
The authors of the current study reviewed the reproductive outcomes and oncological safety of Tamoxifen treatment interruption, which is a timely and relevant topic. The manuscript's structure is also good, demonstrating proper logical flow. The quantitative data for analysis is also strong, showing a 74% pregnancy achievement rate.
Comments:
- Please consider adding primary references in addition to reviews for a few sentences.
- Does the manuscript disclose that it appears largely AI-generated?
Example:
- The authors produced a very informative review. However, it would be beneficial to include a graphical abstract in the manuscript.
- The sections on Emerging Trends and Future Directions could benefit from enhancement. Furthermore, adding a figure for this section will aid the reader.

Author Response
We want to thank the reviewer for the valuable comments, which significantly contributed to enhancing our work, and we hope it will now meet the standards for publication.
Comment 1. Please consider adding primary references in addition to reviews for a few sentences.
Response 1. Thank you, we added primary references in addition to reviews, such as
Sun MR, Steward AC, Sweet EA, Martin AA, Lipinski RJ. Developmental malformations resulting from high-dose maternal tamoxifen exposure in the mouse. PLoS One. 2021 Aug 17;16(8):e0256299. doi: 10.1371/journal.pone.0256299. PMID: 34403436; PMCID: PMC8370643.
Berger JC, Clericuzio CL. Pierre Robin sequence associated with first trimester fetal tamoxifen exposure. Am J Med Genet A. 2008 Aug 15;146A(16):2141-4. doi: 10.1002/ajmg.a.32432. PMID: 18629878.
Shima T, Sasaki Y, Itoh M, Nakashima A, Ishii N, Sugamura K, Saito S. Regulatory T cells are necessary for implantation and maintenance of early pregnancy but not late pregnancy in allogeneic mice. J Reprod Immunol. 2010 Jun;85(2):121-9. doi: 10.1016/j.jri.2010.02.006. Epub 2010 May 2. PMID: 20439117.
Samstein RM, Josefowicz SZ, Arvey A, Treuting PM, Rudensky AY. Extrathymic generation of regulatory T cells in placental mammals mitigates maternal-fetal conflict. Cell. 2012 Jul 6;150(1):29-38. doi: 10.1016/j.cell.2012.05.031. PMID: 22770213; PMCID: PMC3422629.
Comment 2. Does the manuscript disclose that it appears largely AI-generated?
Response 2. I would like to clarify that none of the main text of the manuscript has been generated using AI tools. The content, including the abstract, was entirely written by the authors based on our research and scientific knowledge. We used both ChatGPT and Grammarly, a language editing tool, solely to assist with grammar, spelling checks, syntax editing, and to improve style and clarity. No tools were used to create any part of the manuscript. I would also appreciate it if the reviewer could kindly share the name or details of the tool used for AI detection. Current academic and scientific consensus highlights the limited reliability and validity of AI-detection tools, especially for formal editorial decisions. Several peer-reviewed studies have shown that these tools often suffer from false positives, particularly in well-written or technical text, and may misclassify content simply based on linguistic patterns. Additionally, none of these tools are currently standardized, peer-reviewed, or universally accepted in academic publishing. For example, recent evaluations of major AI-detection systems (such as GPTZero and Turnitin) have shown both false-positive rates as high as 27% in academic abstracts and false-negative rates up to 35%, undermining their utility for definitive judgments. (see references). Please let me know if any further clarification or documentation is needed. I remain at your disposal for any additional information you may require.
However, to be even clearer, we added an AI statement in the manuscript
References: Weber-Wulff, D., Anohina-Naumeca, A., Bjelobaba, S. et al. Testing of detection tools for AI-generated text. Int J Educ Integr 19, 26 (2023). https://doi.org/10.1007/s40979-023-00146-z https://lawlibguides.sandiego.edu/c.php?g=1443311&p=10721367&utm_source=chatgpt.com https://arstechnica.com/information-technology/2023/09/openai-admits-that-ai-writing-detectors-dont-work/?utm_source=chatgpt.com
Popkov AA, Barrett TS. AI vs academia: Experimental study on AI text detectors' accuracy in behavioral health academic writing. Account Res. 2024 Mar 22:1-17. doi: 10.1080/08989621.2024.2331757. Epub ahead of print. PMID: 38516933.
Comment 3. The authors produced a very informative review. However, it would be beneficial to include a graphical abstract in the manuscript.
Response 3. Thank you, we added a new Figure (Figure 2) synthesizing the findings. We will consider adding a graphical abstract if the manuscript is suitable for publication, in a later stage.
Comment 4. The sections on Emerging Trends and Future Directions could benefit from enhancement. Furthermore, adding a figure for this section will aid the reader.
Response 4. We added a new Figure to aid the readers (Figure 2) and improved the section accordingly.
Reviewer 2 Report
Comments and Suggestions for Authors
The review by Maiorano et al summarized the findings of tamoxifen usage and fertility in HR+ BC patients. This topic is important and yet unexplored. Research like this should be encouraged and acknowledged. The format of data representation could be improved.
Here are some minor suggestions for consideration:
- For the Supplementary Table 1, I understand the color of dot means the bias risk level, but only when I read the related paragraph in the manuscript. Please explain the colors in the caption in the SI file.
- In the introduction, please add some information about general treatments for HR+ BC patients, and information about tamoxifen e.g. structure, usage, efficacy in HR+ BC patients. Are there any other treatments are usually used simultaneously with tamoxifen? If so, will these treatments also effect the fertility?
- Please use sub-sections in the results section, either by paper or by key findings. Since there were 3 papers selected, the author could work through the key findings of each paper in each section. And please cite the key figures. Right now there were only text both in the results and discussion sections.
- Please use sub-sections in the discussion section as well. They can be grouped under different topics: teratogenic risks of tamoxifen, oncological safety of tamoxifen interruption, cancer recurrence risk, prospective protection of tamoxifen to ovarian function etc. Add key figures or tables.
- Are there any reports on the biological mechanisms of either the teratogenic effects by tamoxifen or its protection to ovarian functions?
Author Response
We want to thank the reviewer for the valuable comments, which significantly contributed to enhancing our work, and we hope it will now meet the standards for publication.
Comment 1. For the Supplementary Table 1, I understand the color of dot means the bias risk level, but only when I read the related paragraph in the manuscript. Please explain the colors in the caption in the SI file.
Response 1. We added the information in the Supplementary file.
Comment 2. In the introduction, please add some information about general treatments for HR+ BC patients, and information about tamoxifen e.g. structure, usage, efficacy in HR+ BC patients. Are there any other treatments are usually used simultaneously with tamoxifen? If so, will these treatments also effect the fertility?
Response 2. We have expanded the Introduction to include a general but brief overview of standard treatments for HR+ breast cancer in premenopausal women, focusing on the role of tamoxifen, its structure and dual estrogenic/anti-estrogenic mechanism, duration of therapy, and established efficacy based on large-scale clinical trials. Furthermore, we have addressed the Reviewer’s question regarding co-administered therapies in the Discussion, highlighting the use of ovarian function suppression and chemotherapy, and their impact on fertility. These revisions provide a more comprehensive view of the treatment landscape and the multifactorial considerations relevant to fertility preservation in this patient population
Comment 3. Please use sub-sections in the results section, either by paper or by key findings. Since there were 3 papers selected, the author could work through the key findings of each paper in each section. And please cite the key figures. Right now there were only text both in the results and discussion sections.
Response 3. We divided the Results section “per-paper”. See also Response 4.
Comment 4. Please use sub-sections in the discussion section as well. They can be grouped under different topics: teratogenic risks of tamoxifen, oncological safety of tamoxifen interruption, cancer recurrence risk, prospective protection of tamoxifen to ovarian function etc. Add key figures or tables.
Response 4. We revised the Discussion dividing it into different topics. We also added a new Figure (Figure 2) for clarity and to help the Readers by encompassing our study’s main findings
Comment 5. Are there any reports on the biological mechanisms of either the teratogenic effects by tamoxifen or its protection to ovarian functions?
Response 5. The discussion section includes clinical evidence regarding tamoxifen's teratogenic effects, particularly studies by Braems et al. and Schuurman et al. To address the biological mechanisms more clearly, we expanded the text to explain that tamoxifen can cross the placenta and disrupt estrogen signaling during fetal development, potentially leading to congenital anomalies. We also clarified the hypothesized mechanism underlying the observed preservation of ovarian reserve, attributing it to tamoxifen’s selective estrogen receptor modulation and its impact on the hypothalamic-pituitary-gonadal axis. New references about reports and studies, such as
Berger JC, Clericuzio CL. Pierre Robin sequence associated with first trimester fetal tamoxifen exposure. Am J Med Genet A. 2008 Aug 15;146A(16):2141-4. doi: 10.1002/ajmg.a.32432. PMID: 18629878.
Sun MR, Steward AC, Sweet EA, Martin AA, Lipinski RJ. Developmental malformations resulting from high-dose maternal tamoxifen exposure in the mouse. PLoS One. 2021 Aug 17;16(8):e0256299. doi: 10.1371/journal.pone.0256299. PMID: 34403436; PMCID: PMC8370643.
have been added.